# Common Features and Intra-Species Variation of *Cutibacterium modestum* Strains, and Emended Description of the Species

**DOI:** 10.3390/microorganisms9112343

**Published:** 2021-11-12

**Authors:** Itaru Dekio, Ken-ichi Okuda, Masako Nishida, Susumu Hamada-Tsutsumi, Tomo Suzuki, Shigeru Kinoshita, Hiroto Tamura, Kenichiro Ohnuma, Yoshiyuki Murakami, Yuki Kinjo, Akihiko Asahina

**Affiliations:** 1Department of Dermatology, The Jikei University School of Medicine, 3-25-8 Nishi-shinbashi, Minato-ku, Tokyo 105-8461, Japan; akihikoasahina@hotmail.com; 2Seikakai Mildix Skin Clinic, 3rd Floor, 3-98 Senju, Adachi-ku, Tokyo 120-0034, Japan; bow.t@aol.jp; 3Department of Bacteriology, The Jikei University School of Medicine, 3-25-8 Nishi-shinbashi, Minato-ku, Tokyo 105-8461, Japan; okuda-k@jikei.ac.jp (K.-i.O.); ykinjo@jikei.ac.jp (Y.K.); 4Department of Clinical Laboratory, Kobe University Hospital, 7-5-2 Kusunoki-cho, Chuo-ku, Kobe 650-0017, Japan; n11msk25@med.kobe-u.ac.jp (M.N.); onumak@med.kobe-u.ac.jp (K.O.); 5Department of Environmental Bioscience, Meijo University, 1-501 Shiogamaguchi, Tenpaku-ku, Nagoya 468-8502, Japan; tsusumu07@gmail.com (S.H.-T.); hiroto@meijo-u.ac.jp (H.T.); 6Department of Ophthalmology, Kyoto Prefectural University of Medicine, 465 Kajii-cho, Hirokoji-agaru, Kawaramachi-dori, Kamigyo-ku, Kyoto 602-0841, Japan; tomosuzu@koto.kpu-m.ac.jp; 7Department of Ophthalmology, Kyoto City Hospital, 1-2 Higashitakada-cho, Mibu, Nakagyo-ku, Kyoto 604-8845, Japan; 8Department of Frontier Medical Science and Technology for Ophthalmology, Kyoto Prefectural University of Medicine, 465 Kajii-cho, Kamigyo-ku, Kyoto 602-0841, Japan; soy.skinoshi@gmail.com

**Keywords:** *Cutibacterium modestum*, *Propionibacterium humerusii*, new species, human isolate, genome, *16S rRNA* gene, *recA* gene, biochemical analysis, MALDI-TOF mass spectrometry

## Abstract

*Cutibacterium modestum* is a new species coined in 2020 as the fifth species of genus *Cutibacterium*, which includes *Cutibacterium acnes*. The species is predicted as a minor but common member of skin microbiome and includes a group tentatively named as “*Propionibacterium humerusii*”. The description of the species has been provided only with a single strain. To establish the characteristics of *C. modestum* and search for possible disease-related subtypes, we investigated the biochemical characteristics of eight live strains and performed in silico comparison of nine genomes. The common features, which included the morphology of Gram-stain positive short rods, the negativity of phenylalanine arylamidase, and several unique MALDI-TOF MS spectral peaks, were considered useful in laboratory identification. Pairwise comparisons of the genomes by in silico DNA–DNA hybridization showed similarity values of 98.1% or larger, which were far higher than the subspecies cutoff of 79–80%. The *16S rRNA* gene sequences of thirteen isolates and genomes were identical. Their *recA* gene sequences were identical except for two strains, HM-510 (HL037PA2) and Marseille-P5998, which showed unique one-nucleotide polymorphisms. The biochemical features using API kits were slightly different among the isolates but far closer than those of the nearest other species, *C. acnes* and *Cutibacterium namnetense*. Spectra of MALDI-TOF mass spectrometry showed slight differences in the presence of *m*/*z* 10,512 (10 kD chaperonin GroS) and three other peaks, further clustering the eight isolates into three subtypes. These results indicated that these isolates did not separate to form subspecies-level clusters, but subtyping is possible by using *recA* gene sequences or MALDI-TOF mass spectrometry spectra. Moreover, this work has confirmed that a group “*P. humerusii*” is included in *C. modestum*.

## 1. Introduction

Genus *Cutibacterium* was established in 2016 as a result of the reclassification of genus *Propionibacterium* into four genera, which are *Propionibacterium*, *Cutibacterium*, *Acidipropionibacterium*, and *Pseudopropionibacterium* (later corrected to *Arachnia*), based on their whole-genome similarity and niche [1,2]. *Cutibacterium* turned out to be a group of four species that reside on the human skin as their major habitat. *Cutibacterium modestum* was proposed in 2020 as the fifth species of the genus, with strain M12^T^ = JCM 33380^T^ = DSM 109769^T^ as the type strain [3]. The strains and genome entries of this new species are accumulating. A recent database search in July 2021 shows fourteen *16S rRNA* gene sequences of the isolates, with >99% similarities to that of the type strain M12^T^, were deposited to the DDBJ/EMBL/GenBank database [4]. The origins of these isolates are the meibomian gland, skin, bone infection, blood, and a cardiac pacemaker device. These isolates include those described with a tentative species name *‘Propionibacterium humerusii’*, which can be considered to be included in *C. modestum*. In the first description of *‘P. humerusii’* by Butler-Wu et al. [5], the bacterial group is described as consisting of 3% of the human skin microbiome. This triggers speculation that this species can have significance in human health and diseases.

However, little description of *C. modestum* is provided yet, and this causes difficulty on the accurate identification in clinical laboratories. This is not only because this species is not yet widely known, but also because there is an ambiguity in the identification procedure as no concrete description of the species exists to rely on. The currently available description is provided based on the analysis of only one strain [3] and there is no evidence that the description covers most of the strains to be further isolated and analyzed. Therefore, there is a need to establish the species description for clinical identification. The accumulation of the correct species-level identifications in laboratories should link to revealing the species’ role in human health and diseases. To establish the common characteristics of this new species, we collected the existing isolates and investigated their morphological, genomic, biochemical, and matrix-assisted laser desorption/ionization time-of-flight (MALDI-TOF) mass spectrometry (MS) features. In the collection process, we used the registered *16S rRNA* gene and genome sequences in the public nucleotide database as a clue, as these strains are registered with different species names. As a result, we could collect four isolates from Japan and the US. Furthermore, one of the institutions provided three more isolates that were unregistered. On the other hand, a couple of the registered isolates were reported as lost, and a couple of the contacted institutions did not reply.

The recent investigation into *Cutibacterium acnes*, the most famous and well-investigated species of the genus, resulted in a deep understanding of its pathogenicity. The species shows deep intraspecies clustering that forms three groups/subspecies, all of which are unique in cellular shape, genome, biochemical properties, proteome, and pathological characteristics, deserving subspecies definitions [6,7]. For *C. modestum*, it is yet unclear whether this new species forms similar distinct groups that may be related to pathogenicity. Therefore, our study also examines whether such clusterings are present in *C. modestum* by using the live strains and database genomes.

## 2. Materials and Methods

### 2.1. Selection of Bacterial Strains

Strain M12^T^ (= JCM 33380^T^ = DSM 109769^T^) was obtained from the human meibomian gland and previously described as the type strain of *C. modestum* [3]. *C. modestum* strains KB17-24694, KB10376, KB11371, and KB24935 were obtained from Kobe University Hospital. A *C. modestum* strain JK19.3 is one of the strains in *C. acnes* biofilm analysis [8] and was shared from the Jikei University, Department of Bacteriology. Two *C. modestum* strains, HM-510 (HL037PA2) and HM-515 (HL044PA1), with registrations as *Propionibacterium acnes*, were obtained from BEI Resources (Manassas, VA, USA). Strains JCM 6425^T^ (*C. acnes* subsp. *acnes*), JCM 6473^T^ (*C. acnes* subsp. *defendens*), and JCM 18919^T^ (*C. acnes* subsp. *elongatum*) were purchased from RIKEN BioResource Research Center Microbe Division/Japan Collection of Microorganisms (Tsukuba, Ibaraki, Japan). A *C. namnetense* strain DSM 29427^T^ was purchased from the Leibniz Institute DSMZ-German Collection of Microorganisms and Cell Cultures (Braunschweig, Germany). The details of these strains are listed in Table 1.

### 2.2. Morphological Assessments

The strains used in this study were grown on trypticase soy agar supplemented with 5% sheep blood (BD, Franklin Lakes, NJ, USA) for 5 days at 37 °C under anaerobic conditions created in an anaerobic jar with an AnaeroPack sachet (Mitsubishi Gas Chemical, Tokyo, Japan). The colonies were observed, and Gram stain was performed using Gram Stain Kit (Stabilized) (BD).

### 2.3. Selection of Genome Data

Genome data for eight strains, M12^T^, HM-510 (HL037PA2), HM-515 (HL044PA1), P08, T33958, Marseille-P5998, HL037PA3, and F0672 were selected for analysis. These, except for M12^T^, were found by searching matches in the NCBI tblastn web tool for whole-genome shotgun contigs (wgs) with *16S rRNA* gene sequence of M12^T^ (accession number LC466959).

In addition, full genome analysis was performed for strain KB17-24694. First, the genomic DNA of the strain was extracted and purified by using Genomic-tip 100/G kit (Qiagen, Hilden, Germany). The concentration and quality of the extracted DNA was checked by using Synergy LX (BioTek, Winooski, VT, USA), QuantiFluor dsDNA System (Promega, Madison, WI, USA), and 5200 Fragment Analyzer System with Agilent HS Genomic DNA 50 kb Kit (Agilent Technologies, Santa Clara, CA, USA), and the material passed for further analysis. For short reads, a library of circularized DNA was created by using MGIEasy FS DNA Library Prep Set and MGIEasy Circularization Kit (MGI Tech, Shenzhen, China). Then, sequencing was performed by using DNBSEQ-G400 (MGI Tech) with a parameter of paired-end 200 bp. For long reads, a library was created by using Ligation Sequence Kit (Oxford Nanopore Technologies, Oxford, UK). Then, sequencing was performed by using GridION (Oxford Nanopore Technologies) with standard procedures. The short reads were removed their adapter sequences and reads with smaller than 127 bp or below Q20 were discarded by using the Cutadapt and Sickle softwares. On the other hand, the long reads were removed their adapter sequences and reads with smaller than 1000 bp were discarded by using the Porechop and Filtlong softwares. The filtered short and long reads were assembled by using Unicycler ver. 0.4.7 to create one chromosome sequence and one plasmid sequence.

### 2.4. Genome Analysis

Genome data for nine strains, M12^T^, HM-510 (HL037PA2), HM-515 (HL044PA1), P08, T33958, Marseille-P5998, HL037PA3, F0672, and KB17-24694 were used (Table 1). In silico DNA–DNA hybridization (digital DDH; dDDH) values of the genomes were calculated in pairwise manner using GBDP under the algorithm “trimming” and 100 replicates of distance formula *d*_5_. These intergenomic distances were used to create a balanced minimum evolution tree with branch support. This analysis was performed by using the TYGS online platform (https://tygs.dsmz.de/, accessed on 8 July 2021) [9].

### 2.5. 16S rRNA Gene and recA Gene Analyses

Nearly full sequences of *16S rRNA* gene of four strains, KB17-24694, KB10376, KB11371, and KB24935, were determined as previously reported [16]. On the other hand, *16S rRNA* gene sequences of strains HM-510 (HL037PA2), HM-515 (HL044PA1), P08, T33958, Marseille-P5998, HL037PA3, and F0672 were extracted from their genome data by using NCBI tblastn web tool (https://blast.ncbi.nlm.nih.gov/Blast.cgi, accessed on 3 July 2021). Furthermore, nearly full sequences of *recA* gene of six strains, KB17-24694, KB10376, KB11371, KB24935, JK19.3, and HM-510 (HL037PA2) were determined by using a newly developed set of primers as below: Cmod_recA_1F 5′ ATGGCAGTGACCGCTGAC 3′, Cmod_recA_480F 5′ GGGTGACTCCCATGTCGGTTT 3′, Cmod_recA_550R 5′ TCAACGCACCGGTCATCTTGCGCA 3′, and Cmod_recA_1046R 5′ CAGAATTCCACCTCACCAGTCTG 3′. PCR amplification using Cmod_recA_1F and Cmod_recA_1046R were performed with the following program: 95 °C for 5 min, followed by 35 cycles consisting of 95 °C for 1 min, 52 °C for 30 s, 68 °C for 1 min, and a final extension period of 68 °C for 5 min, giving a 1046 bp PCR product. In addition, *recA* gene sequences of strains M12^T^, HM-510 (HL037PA2), HM-515 (HL044PA1), P08, T33958, Marseille-P5998, HL037PA3, and F0672 were extracted from their genome data by using NCBI tblastn web tool.

The *16S rRNA* gene sequences were prepared as a single text file in FASTA format. The sequence file was pasted in a DNA sequence alignment mode of MEGA X ver.10.2.5 software [17] and aligned by MUSCLE alignment mode. The *recA* gene sequences were also aligned in the same manner.

### 2.6. Biochemical Profiling

*C. modestum* strains KB17-24694, KB10376, KB11371, KB24935, JK19.3, and *C. namnetense* strain NTS 31307302^T^ were cultured anaerobically for five days as described in Section 2.2. Using API kits (API Coryne and API Rapid ID 20A, bioMérieux, Marcy-l’Étoile, France), biochemical analysis was performed according to the manufacturer’s instructions for these strains. In addition, catalase tests were performed by using API Coryne kit for *C. modestum* strain M12^T^ and all three *C. acnes* strains.

### 2.7. MALDI-TOF Mass Spectrometry

MALDI-TOF mass spectrometry profiles of the strains were obtained as reported by Dekio et al. [18] with the following modifications. The spectra were obtained with a MALDI-TOF MS Autoflex Speed Biotyper (Bruker, Billerica, MA, USA) with parameters as follows: mass range, 2000–20,000; ion source 1, 19.50 kV and 2, 18.50 kV; lens, 6 kV; detector gain voltage, 2600 V (linear base). The spectra were obtained at least in triplicate and the representative spectrum with the clearest and richest pattern was selected for each strain. The raw spectral data was analysed with flexAnalysis v3.4 software (Bruker). The raw data in mzXML format was deposited to jPOST Repository, with accession number JPST001214 (PXID: PXD026667) (URL: https://repository.jpostdb.org/, accessed on 21 October 2021).

To assign group-specific peaks, genome data with annotations for strains M12^T^ (BJEN01), KB17-24694 (AP024747), and HM-515 (HL044PA1) (ADZU01) were analyzed. Using these genome data, protein molecular weight lists based on genome comparisons were created as described previously [18].

## 3. Results

### 3.1. Morphology

After five days of anaerobic culture, all eight strains showed very similar white, opaque, and lenticular colonies with diameters of 0.9–1.0 mm. Gram stain revealed all strains being Gram-positive short rods (Figure 1).

### 3.2. Genome Analysis

The dDDH values for the pairwise comparisons of *C. modestum* isolates were presented in Table 2. These values ranged between 98.1% and 100%, far above those not only of the species cutoff of 70% [19] but also of the subspecies cutoff of 79–80% [20] (green highlight, Table 2). It was a striking contrast to the pairwise comparisons of the three *C. acnes* genomes, which showed the values between 75.5% and 73.1%, reflecting their subspecies-level differences (yellow highlight, Table 2). On the other hand, the values between *C. modestum* genomes and those of other *Cutibacterium* species were between 22.3% and 32.3% with all the other type strains, which were below the species cutoff of 70%.

The value between the genomes of HM-510 (HL037PA2) and HL037PA3 was 100%, which was the only 100% among the pairwise comparisons. Considering the very similar characteristics of these two genomes (both 2.61 Mb and G + C content 59.94%), and the similarity of these strain names, we speculated these were identical. Based on these findings, we excluded HL037PA3 for further analysis.

A phylogenetic tree by using these dDDH values was successfully created (Figure 2). The three subspecies of *C. acnes* branched deep, while genomes of *C. modestum* did not.

### 3.3. 16S rRNA Gene and recA Gene Analyses

Nearly full *16S rRNA* gene sequences of four KB isolates were successfully determined. The comparison across the twelve *16S rRNA* sequences (Table 1, excluding HL037PA3) revealed that all these sequences were identical.

Also, nearly full *recA* gene sequences of the five KB and JK isolates were successfully determined. The comparison across the twelve *recA* gene sequences (Table 1, excluding HL037PA3) revealed that two polymorphisms exist, which were G57A for strain HM-510 (HL037PA2) and T66C for strain Marseille-P5998 (Figure 3). We additionally determined the gene sequence for strain HM-510 (HL037PA2) to confirm this polymorphism. These polymorphisms in the coding region did not affect the translation to amino acids, which are glutamic acid (E) for frame 55–57 (GAG and GAA) and histidine (H) for frame 64–66 (CAT and CAC).

### 3.4. Biochemical Analysis

The common features across these twelve isolates were: positive for catalase and arginine arylamidase, and negative for urease, α-galactosidase, β-galactosidase, β-galactosidase-6-phosphate, β-glucosidase, α-arabinosidase, mannose fermentation, raffinose fermentation, glutamate decarboxylase, α-fucosidase, alkaline phosphatase, leucylglycine arylamidase, pyroglutamic acid arylamidase, tyrosine arylamidase, glutamyl glutamic acid arylamidase, β-glucuronidase, aesculin hydrolysis (β-glucosidase), xylose fermentation, mannitol fermentation, maltose fermentation, lactose fermentation, sucrose fermentation, and glycogen fermentation. On the other hand, different reactions were observed among these twelve strains for arginine dihydrolase, α-glucosidase, *N*-acetyl-β-glucosaminidase, nitrate, indole, proline arylamidase, phenylalanine arylamidase, leucine arylamidase, alanine arylamidase, glycine arylamidase, histidine arylamidase, serine arylamidase, pyrazinamidase, pyrrolidonyl arylamidase, β-galactosidase, gelatin hydrolysis, glucose fermentation, and ribose fermentation (Table 3).

The characteristic feature of *C. modestum* strains was a marked negativity among various tests, including α-glucosidase, phenylalanine arylamidase, leucine arylamidase, alanine arylamidase, histidine arylamidase, serine arylamidase, pyrazinamidase, β-galactosidase, gelatin hydrolysis, and ribose fermentation tests. These results indicated relatively lower reactivity of *C. modestum* strains compared with the type strains of three *C. acnes* subspecies and *C. namnetense*. Especially, the negativity of phenylalanine arylamidase was considered to discriminate *C. modestum* from the other two species. *N*-Acetyl-β-glucosaminidase and indole tests were negative for *C. modestum* type strain M12^T^ but were positive for most other *C. modestum* strains.

### 3.5. MALDI-TOF Mass Spectrometry

MALDI-TOF MS spectra of all eight strains of *C. modestum* and type strains of *C. acnes* and *C. namnetense* were successfully obtained (Figure 4). The spectra of *C. modestum* comprised of three groups, namely, Group A, B, and C. Group A included five strains, which were strains M12^T^, KB10376, KB24935, JK19.3, and HM-510 (HL037PA2). Group B, which included peaks of *m*/*z* (= *x*-axis) 4957, 5016, 9914, and 10,034, was produced from strains KB17-24694 and KB11371. Peaks with *m*/*z* values of 4957 and 5016 were estimated as doubly charged ions ([M + 2H]^2+^) and *m*/*z* 9914 and 10,034 were singly charged ions ([M + H]^+^) of the same molecules (probably proteins), namely, molecules X and Y. Group C, which included peaks with *m*/*z* values of 5337 and 10,674, was produced from strain HM-515 (HL044PA1). In the same manner, *m*/*z* 5337 was a doubly charged ion ([M + 2H]^2+^) and *m*/*z* 10,674 was a singly charged ion ([M + H]^+^) of the same molecule, namely, molecule Z.

We tried to identify molecules X, Y, and Z by comparing the genomes of strains of Groups A (strain M12^T^: BJEN01), B (strain KB17-24694: AP024747), and C (strain HM-515: ADZU01). However, it was not possible for these molecules because there were no gene sequence in Group B and C genomes that corresponded to proteins with the appropriate molecular weights and matching their observed mass to charge ratio (*m*/*z*). Following this result, possible post-translational modifications were considered. The observed *m*/*z* of the target molecule should increase 14, 42, or 56 by methylation, acetylation, and propionylation, respectively. As a result, the candidate proteins with modifications for these ions were suggested as follows: the candidates for molecule X include C25_01851 hypothetical protein (molecular weight = 10,026.17, *m*/*z* -Met [M + H]^+^ = 9895.99, after one methylation, 9909), C25_00492 hypothetical protein (molecular weight = 9869.97, *m*/*z* [M + H]^+^ = 9870.98, after one acetylation or three methylations, 9911), C25_01631 hypothetical protein (molecular weight = 10,041.49, *m*/*z* -Met [M + H]^+^ = 9911.31), and C25_00877 phosphoribosyl-ATP pyrophosphatase (molecular weight = 10,028.26, *m*/*z* -Met [M + H]^+^ = 9898.08, after one methylation, 9912), all coded in the KB17-24694 genome. The candidates for molecule Y include C25_00199 hypothetical protein (molecular weight = 10,014.30, *m*/*z* [M + H]^+^ = 10,015.31, after one methylation, 10,028), C25_00350 hypothetical protein (molecular weight = 10,135.39, *m*/*z* -Met [M + H]^+^ = 10,005.21, after two methylations, 10,033), and C25_00245 hypothetical protein (molecular weight = 10,124.87, *m*/*z* -Met [M + H]^+^ = 9994.69, after one acetylation or three methylations, 10,036), all coded in the KB17-24694 genome. The candidates for molecule Z include WP_002529140.1_1400 YbaB/EbfC family nucleoid-associated protein (molecular weight = 10,785.98, *m*/*z* -Met [M + H]^+^ = 10,655.80, after one methylation, 10,669), WP_172623584.1_1907 hypothetical protein (molecular weight = 10,765.11, *m*/*z* -Met [M + H]^+^ = 10,634.93, after one acetylation or three methylations, 10,676), and WP_002528091.1_984 ATP-binding cassette domain-containing protein (molecular weight = 10,751.39, *m*/*z* -Met [M + H]^+^ = 10,621.21, after either one propionylation, one acetylation plus one methylation, or four methylations, 10,677), all coded in the HM-515 (HL044PA1) genome.

## 4. Discussion

Our phenotypic, genomic, biochemical, and MALDI-TOF MS analyses revealed the relative uniformity among *C. modestum* strains. These strains and genomes did not form clusters at the subspecies level but exhibited differences at the lower level in *recA* gene, biochemical, and MALDI-TOF MS spectral characteristics, suggesting the feasibility of strain-level typings.

### 4.1. Common Characteristics of Cutibacterium modestum

Our results outlined the description in the previous new species proposal [3], and these did not contradict the description of genus *Cutibacterium* [1] nor that of previous genus *Propionibacterium* [21]. Furthermore, the results indicate the taxonomical position of the type strain M12^T^ being a typical strain among *C. modestum* strains. The results of this strain include the genome being located very close to other strains in the phylogenetic tree (Figure 2), the identical *recA* sequence to most other strains (Figure 3), the typical biochemical behavior except negativity in *N*-acetyl-β-glucosaminidase and indole tests (Table 3), and a common Group A MALDI-TOF MS spectrum (Figure 4).

In the biochemical analysis by using API kits, the numbers of the positive reactions by *C. modestum* strains were the smallest among the three *Cutibacterium* species (Table 3), verifying the modest nature of this species, as in its epithet. The results of *C. modestum* were the most similar to those of *C. acnes* subsp. *elongatum*. Among the 32 reactions that consist of the kits, the negativity of phenylalanine arylamidase is specific to distinguish *C. modestum* from *C. acnes* or *C. namnetense*. Moreover, the negativity of α-glucosidase, leucine arylamidase, β-galactosidase, gelatin hydrolysis, and ribose fermentation are key reactions of distinguishing *C. modestum*.

On the other hand, MALDI-TOF MS spectra of *C. modestum* showed distinct patterns that can be discriminated from those of the nearest two species as shown in Figure 4, and this uniqueness suggests the strength of the MALDI-TOF MS technique for *Cutibacterium* species-level identification. The key prominent peaks in the spectra were 4765, 5256, 5857, 6987, 7424, 9530, 10,510, and 11,713, the *m*/*z* value (*x*-axis) of which are based on genome-based calculations. All these peaks reflect intracellular proteins, which are DNA-binding protein HU [M + 2H]^2+^, 10 kD chaperonin GroS -Val [M + 2H]^2+^, Lsr2-like protein -Met [M + 2H]^2+^, 7 kD antitoxin -Met [M + H]^+^, CsbD-like protein -Val/-Met [M + H]^+^, DNA-binding protein HU [M + H]^+^, 10 kD chaperonin GroS -Val [M + H]^+^, and Lsr2-like protein -Met [M + H]^+^, respectively, as reported in our previous work of an interspecies comparative proteogenomics [18]. The majority of these peaks were identical among *C. modestum* strains, reflecting the uniformity of corresponding coding genes. This fact is strikingly different from *C. acnes* subtypes, which exhibit different characteristic spectra [18]. The comparative proteogenomics also raised the possibility of the post-translational modification of the observed proteins in the spectra.

The above results of the biochemical and MALDI-TOF MS analysis should be useful for species-level identification in clinical laboratories.

### 4.2. Variation of Cutibacterium modestum

We sought for subspecies-level discrimination at first, in analogy with *C. acnes*, which includes three subspecies that are different in various aspects, that is in morphology, genome, *16S rRNA* gene, *recA* gene, biochemical tests, and pathogenicity [4]. However, the *C. modestum* isolates did not show differences in this level. According to our pairwise genome comparisons (dDDH values), the variation range of *C. modestum* isolates was relatively small to repel subspecies discrimination (Table 2). The lack of heterogeneity in *16S rRNA* gene sequences is consistent with this observation. Following this, we established a method for *recA* gene typing to investigate the possibility of housekeeping gene typing of this relatively homogeneous population. As a result, we were able to observe three groups based on one-nucleotide polymorphisms: the first group including strain M12^T^ and the other two groups with mutations G57A and T66C. Although the polymorphisms we observed were within a single nucleotide, we consider this to be significant. This is because a similar one-base difference can distinguish *C. acnes* type IA and IB, which are different in pathogenicity [22]. In a case of *C. acnes*, *recA* gene typing [22] has advanced to multilocus sequence typing (MLST) [23] and to subspecies proposal [6,7], both of which link to the pathogenicity of distinguished groups. Our *recA* gene typing may become the first step in the further investigation to understand the niche and the role of this new species.

The biochemical analysis revealed slight differences among *C. modestum* strains. The prominent differences were the negative results for arginine dihydrolase in strains M12^T^ and KB24935, for *N*-acetyl-β-glucosaminidase in strain M12^T^, for proline arylamidase in strain KB17-24694, and for glucose fermentation in strain HM-510 (HL037PA2). The glucose fermentation negativity in strain HM-510 (HL037PA2) suggests the link of this result to *recA* gene polymorphism G57A, but further accumulation of isolates is needed for verification.

Mixed results were observed for the presence of indole, which indicates that indole detection by using API Rapid ID 32A kit is not useful for *C. modestum* identification. This is also the case for *C. acnes*; the product datasheet of this kit states that the positive rate of indole reaction for *C. acnes* is only 62%, although Bergey’s manual states all three subspecies of *C. acnes* are indole positive [21]. Goldenberger et al. emphasizes the importance of the indole test along with the catalase test with regard to species delineation. In the article, the need for clarification of strain M12^T^, which was negative for indole, was hinted with reference to their isolate 602588-20-USB and strain P08, which were positive [24]. Our results of eight *C. modestum* strains were weakly positive in six strains and negative in two strains, and no strong positives were observed. However, our genome analysis indicated that strain P08, the indole positive strain, is included in *C. modestum* (Table 2, Figure 2). Moreover, both positive and negative strains share the same *recA* gene sequence (Figure 3) and MALDI-TOF MS spectra (Group A in Figure 4). To investigate whether the difference among strains is derived from the presence/absence of tryptophanase gene, we checked the presence of tryptophanase gene in the four *C. modestum* genomes of the isolates used in our biochemical analysis (strain M12^T^, BJEN01; strain KB17-24694, AP024747; strain HM-510, ADYH01; strain HM-515, ADZU01). All genomes possessed one each of the gene, with 46–49% similarity from *Escherichia coli* tryptophanase gene (*tnaA*) and 87% similarity from the correspondent gene in *C. acnes*. Considering all these results, we speculate that negative, weak positive, and clear positive results for indole can be obtained from *C. modestum* strains, although the tryptophanase gene is universal.

MALDI-TOF MS spectra of *C. modestum* showed three spectral patterns, the finding of which proves to be reliable to perform easy and cost-effective type analysis. Unfortunately, the MALDI-TOF MS spectra typings were not congruent with any of the *recA* gene or biochemical results. More analysis is needed to establish a meaningful typing scheme explaining their differences in niche and pathogenesis.

### 4.3. “Propionibacterium humerusii” Is Included in Cutibacterium modestum

The presence of *C. modestum* was initially suggested with a tentative name as *“Propionibacterium humerusii”* in 2011 with genomic analysis of an isolate P08 [5]. Unfortunately, this work did not result in the formal proposal of the species. Our proposal of *C. modestum* followed, and at that stage, we were aware of the 16S rRNA gene sequence of strain P08 being identical to the type strain of *C. modestum* [3]. Very recently, Goldenberger et al. stated that *C. modestum* and *“P. humerusii”* represent the same species [24]. However, we consider it differently. In the strict sense, the range that *“P. humerusii”* covers has not been defined, and thus it is unclear whether *“P. humerusii”* covers all strains of *C. modestum*. Our conclusion based on this history is that *C. modestum* is not equal to *“P. humerusii”*, but the former includes the latter.

### 4.4. Emended Description of Cutibacterium modestum Dekio et al., 2020

The description is given as before [3] with the following correction. Using API Rapid ID 32A and API Coryne kits, positive reactions are observed for catalase and arginine arylamidase; negative reactions are observed for urease, α-galactosidase, β-galactosidase, β-galactosidase-6-phosphate, α-glucosidase, β-glucosidase, α-arabinosidase, mannose fermentation, raffinose fermentation, glutamate decarboxylase, α-fucosidase, alkaline phosphatase, leucylglycine arylamidase, phenylalanine arylamidase, leucine arylamidase, pyroglutamic acid arylamidase, tyrosine arylamidase, alanine arylamidase, histidine arylamidase, glutamyl glutamic acid arylamidase, serine arylamidase, pyrazinamidase,β-glucuronidase, β-galactosidase, aesculin hydrolysis (β-glucosidase), xylose fermentation, mannitol fermentation, maltose fermentation, lactose fermentation, sucrose fermentation, and glycogen fermentation. Prominent mass ion peaks obtained by MALDI-TOF mass spectrometry are 3493, 3712, 4765, 5256, 5857, 6987, 7424, 9530, 10,510, and 11,713.

## 5. Conclusions

As described above, our morphological, biochemical, and MALDI-TOF MS analyses outlined the general features of *C. modestum*, which are useful in species-level diagnostic identification. All *C. modestum* isolates were Gram-stain positive short rods. Negativity of phenylalanine arylamidase should be a key to distinguish *C. modestum* from *C. acnes* and *C. namnetense*. MALDI-TOF MS spectra of *C. modestum* is unique and useful for species identification once the spectral data is registered in the identification system.

Moreover, multidimensional analysis of the strains and genomes of *C. modestum* revealed that strain-level (below subspecies-level) typings of *C. modestum* are possible by using *recA* gene sequences, biochemical reactions, or MALDI-TOF MS spectra. The G57A polymorphism of *recA* gene is especially worth attention, as this is linked to a negative result in the glucose fermentation test in one strain. On the other hand, these strains and genomes did not form subspecies-level clusters. To note, however, this study does not exclude the possibility of future discovery of the subspecies-level clusterings by further accumulation of isolates.

## Figures and Tables

**Figure 1 microorganisms-09-02343-f001:**
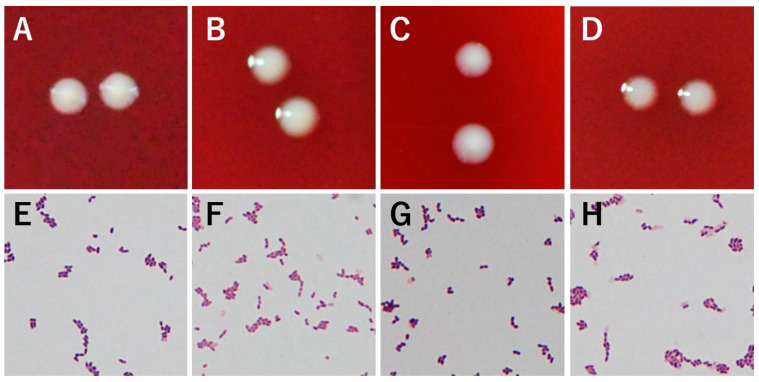
Colony and Gram stain appearances of four strains of *C. modestum*. (**A**–**D**): Colonies of strains, M12^T^ = JCM 33380^T^, KB17-24694, JK19.3, and HM-510 (HL037PA2), respectively. (**E**–**H**): Gram stain appearances of the same strains. These appearances were very similar across the strains.

**Figure 2 microorganisms-09-02343-f002:**
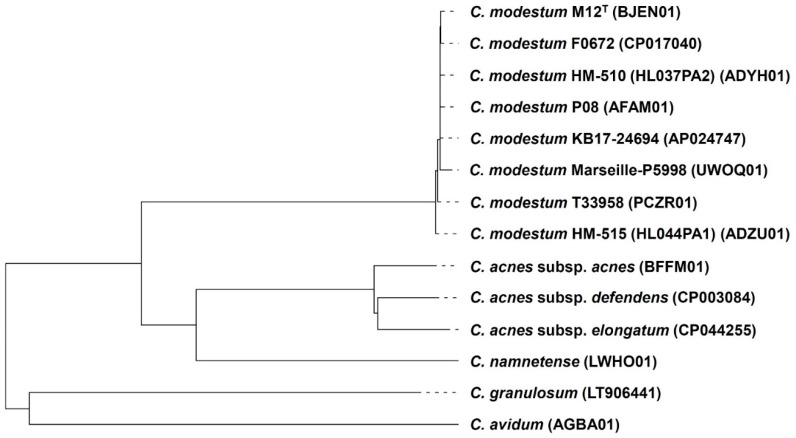
Phylogenetic tree of nine *C. modestum* isolates and type strains of all other species and subspecies of genus *Cutibacterium* based on their genome sequences created by using TYGS platform. *C. modestum* genomes other than M12^T^ and KB17-24694 were registered with other species names but estimated as *C. modestum* with >99% similarities of *16S rRNA* gene sequences with M12^T^.

**Figure 3 microorganisms-09-02343-f003:**
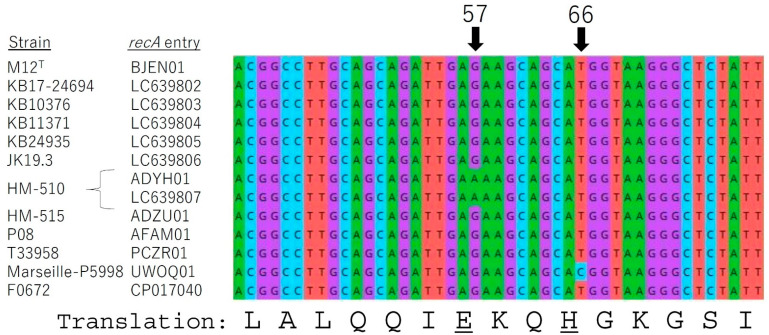
Comparison of *recA* gene sequences of *C. modestum* strains by using MEGA X ver.10.2.5 software. As the sequence obtained from the genome of strain HM-510 (HL037PA2) (accession number ADYH01) had a polymorphism G57A, the gene was amplified for confirmation (LC639807). Polymorphisms G57A and T66C were observed as the only polymorphisms in the almost-full gene sequences. Amino acid abbreviations derived from nucleotide frames including polymorphisms were underlined. These polymorphisms did not affect the amino acid translations.

**Figure 4 microorganisms-09-02343-f004:**
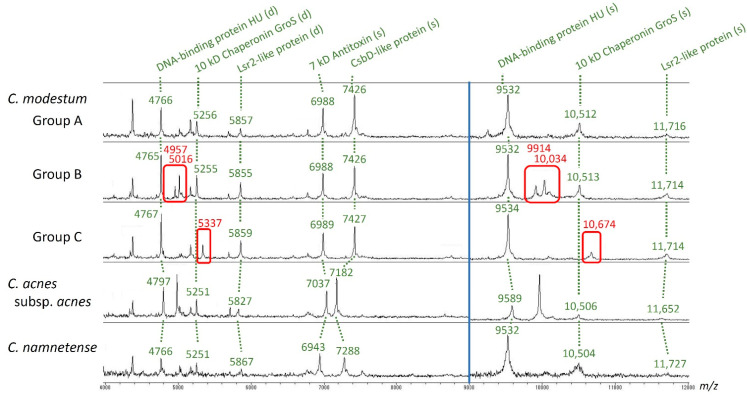
MALDI-TOF MS spectra of *C. modestum* strains and the type strains of the nearest two species. The spectra of the eight strains formed three groups, Group A (strains M12^T^, KB10376, KB24935, JK19.3, HM-510 (HL037PA2)), Group B (KB17-24694, KB11371), and Group C (HM-515 (HL044PA1)). The *y*-axis is different between 4000–9000 *m*/*z* and 9000–12,000 *m*/*z* ranges. The peak assignments to proteins were obtained from a previous report [18]. (s), (d): Singly and doubly charged ions of the protein. Differences between Groups were highlighted in red.

**Table 1 microorganisms-09-02343-t001:** List of bacterial isolates used in this study. These isolates include those registered with other species name. T indicates type strains of the species/subspecies. Brackets: part of genome sequence extracted for analysis. * Newly obtained sequences in this work.

Species and Strain Name	Origin	Country (Isolation Year)	Genome	*16S rRNA*	*recA*	Reference
*Cutibacterium modestum*						
(Isolates, gene sequences, and genome data used in this study)						
M12^T^ = JCM 33380^T^	meibomian gland	Japan (2013)	BJEN01	LC466959	(BJEN01)	[3]
KB17-24694	vertebral infection	Japan (2017)	AP024747 *	LC637867 *	LC639802 *	
KB10376	blood	Japan (2019)	NA	LC637868 *	LC639803 *	
KB11371	blood	Japan (2019)	NA	LC637869 *	LC639804 *	
KB24935	cerebrospinal fluid	Japan (2020)	NA	LC637870 *	LC639805 *	
JK19.3	cardiac pacemaker device	Japan	NA	LC341281	LC639806 *	[8]
HM-510 (HL037PA2)	skin	US	ADYH01	(ADYH01)	LC639807 * /(ADYH01)	[5,10,11]
HM-515 (HL044PA1)	skin	US	ADZU01	(ADZU01)	(ADZU01)	[5,10,11]
(Only genome data used in this study)						
P08	humeral bone (infected)	US	AFAM01	(AFAM01)	(AFAM01)	[5,10]
T33958	skin (wound)	US	PCZR01	(PCZR01)	(PCZR01)	
Marseille-P5998	vagina	France	UWOQ0	(UWOQ01)	(UWOQ01)	
HL037PA3	skin	US	ADXV01	(ADXV01)	(ADXV01)	[5,10,11]
F0672	oral cavity	US	CP017040	(CP017040)	(CP017040)	
*Cutibacterium acnes* subsp. *acnes*						
JCM 6425^T^ = ATCC 6919^T^	skin (acne lesion)	UK	CP044255			[6,7,12,13]
*Cutibacterium acnes* subsp. *defendens*						
JCM 6473^T^ = ATCC 11828^T^	subcutaneous abscess	unknown	CP003084			[7,12,13]
*Cutibacterium acnes* subsp. *elongatum*						
JCM 18919^T^	skin	Japan	BFFM01			[6,14]
*Cutibacterium namnetense*						
NTS 31307302^T^ = DSM 29427^T^	tibial bone (infected)	France	LHWO01			[12,13,15]
*Cutibacterium avidum*						
NCTC 11864^T^ = ATCC 25577^T^	unknown	unknown	AGBA01			
*Cutibacterium granulosum*						
NCTC 11865^T^	unknown	unknown	LT906441			

**Table 2 microorganisms-09-02343-t002:** In silico DDH (dDDH) values. 1. *Cutibacterium modestum* M12^T^ = JCM 33380^T^, 2. KB17-24694, 3. HM-510 (HL037PA2), 4. HM-515 (HL044PA1), 5. P08, 6. T33958, 7. Marseille-P5998, 8. F0672, 9. *Cutibacterium acnes* subsp. *acnes* JCM 6425^T^, 10. *Cutibacterium acnes* subsp. *defendens* JCM 6473^T^, 11. *Cutibacterium acnes* subsp. *elongatum* JCM 18919^T^, 12. *Cutibacterium namnetense* NTS 31307302^T^, 13. *Cutibacterium avidum* ATCC 25577^T^, 14. *Cutibacterium granulosum* NCTC 11865^T^. dDDH values >70% indicates the same species, DDH value >79% indicates the same subspecies. Highlights in green and yellow indicate dDDH values >79% and between 79% and 70%, respectively.

Strain (Genome Entry)		1	2	3	4	5	6	7	8	9	10	11	12	13	14
1	BJEN01	100	99.9	99.7	99.7	99.8	99.7	98.2	99.9	31.6	32.3	31.4	30.3	23.0	22.3
2	AP024747		100	99.8	99.9	99.9	99.9	98.5	99.9	31.4	32.1	31.4	30.3	23.1	22.5
3	ADYH01			100	99.7	99.7	99.7	98.2	99.8	31.6	32.3	31.3	30.3	23.1	22.2
4	ADZU01				100	99.7	99.8	98.1	99.9	31.6	32.3	31.4	30.3	23.1	23.1
5	AFAM01					100	99.8	98.2	99.9	31.6	32.3	31.4	30.3	23.1	22.3
6	PCZR01						100	98.2	99.9	31.6	32.3	31.4	30.3	23.1	23.0
7	UWOQ01							100	98.5	31.5	32.2	31.4	30.3	23.0	22.1
8	CP017040								100	31.4	32.1	31.4	30.3	23.1	22.5
9	CP044255									100	75.5	73.1	36.0	23.6	22.5
10	CP003084										100	74.1	35.3	23.5	22.2
11	BFFM01											100	35.4	23.3	21.7
12	LWHO01												100	23.7	21.8
13	AGBA01													100	24.0
14	LT906441														100

**Table 3 microorganisms-09-02343-t003:** Summary of biochemical analysis using API. 1–8 are strains of *C. modestum* and 9–12 are those of other *Cutibacterium* species. 1. M12^T^ = JCM 33380^T^, 2. KB17-24694, 3. KB10376, 4. KB11371, 5. KB24935, 6. JK19.3, 7. HM-510 (HL037PA2), 8. HM-515 (HL044PA1), 9. *Cutibacterium acnes* subsp. *acnes* JCM 6425^T^, 10. *Cutibacterium acnes* subsp. *defendens* JCM 6473^T^, 11. *Cutibacterium acnes* subsp. *elongatum* JCM 18919^T^, 12. *Cutibacterium namnetense* NTS 31307302^T^. Identical results among all strains are not shown (see 3. Results, 3.4. Biochemical analysis section). +, positive; −, negative; w, weak. All data are from this study except for M12^T^ and *C. acnes* strains, which are cited from [3] with corrections based on the original results.

Test	1	2	3	4	5	6	7	8	9	10	11	12
API Rapid ID 32A												
Arginine dihydrolase	−	w	w	w	−	w	w	w	−	−	−	−
α-Glucosidase	−	−	−	−	−	−	−	−	−	+	−	+
N-Acetyl-β-glucosaminidase	−	+	+	+	+	+	+	w	+	+	+	+
Nitrate	+	+	w	+	+	+	+	+	+	+	+	+
Indole	−	w	w	−	w	w	w	w	−	w	−	+
Proline arylamidase	+	−	+	+	+	+	+	+	+	+	+	+
Phenylalanine arylamidase	−	−	−	−	−	−	−	−	w	+	+	w
Leucine arylamidase	−	−	−	−	−	−	−	−	−	+	−	w
Alanine arylamidase	−	−	−	−	−	−	−	−	+	+	−	+
Glycine arylamidase	w	−	w	w	w	w	w	w	+	+	w	+
Histidine arylamidase	−	−	−	−	−	−	−	−	−	−	−	w
Serine arylamidase	−	−	−	−	−	−	−	−	+	+	−	+
API Coryne												
Pyrazinamidase	−	−	−	−	−	−	−	−	−	−	−	+
Pyrrolidonyl arylamidase	w	w	w	w	−	−	−	−	w	+	+	−
β-Galactosidase	−	−	−	−	−	−	−	−	+	+	−	+
Gelatin	−	−	−	−	−	−	−	−	+	+	−	+
Glucose	+	+	+	+	+	+	−	+	+	+	w	+
Ribose	−	−	−	−	−	−	−	−	+	+	−	+

## Data Availability

The genome sequence for *Cutibacterium modestum* strain KB17-24694 is openly available in the DDBJ-EMBL/GenBank database, accession numbers AP024747 (a circular chromosome) and AP024748 (a linear plasmid). *16S rRNA* gene sequence (almost full gene) for strains KB17-24694, KB10376, KB11371, and KB24935 are openly available in the DDBJ-EMBL/GenBank database, accession numbers LC637867–LC637870. *recA* gene sequence (almost full gene) for strains KB17-24694, KB10376, KB11371, KB24935, JK19.3 and HM-510 (HL037PA2) are openly available in the DDBJ-EMBL/GenBank database, accession number LC639802–LC639807. The MALDI-TOF MS spectra data presented in this study is openly available in the jPOST Repository [https://repository.jpostdb.org/, accessed on 21 October 2021], reference number JPST001214 (PXID: PXD026667).

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
