# Peer review of "Common Features and Intra-Species Variation of Cutibacterium modestum Strains, and Emended Description of the Species"

_microorganisms, 2021, doi:10.3390/microorganisms9112343_

Round 1
Reviewer 1 Report
The authors demonstrated a low heterogeneity among isolates of Cutibacterium modestum emended the description of the species.
The title should be: Low heterogeneity among isolates of Cutibacterium modestum, a resident of human skin with a possible infectious nature, and emended description of the species. There is only one description.
The value of this study is not so much in the emended description of the species but in criteria for strain identification useful in diagnostic. This should be clearly stated in the abstract.
The emendation of the species is in this form not valid. It has to follow the taxonomic rules. See e. g. Yaneechoutte et al., Int J Syst Evol Microbiol 2019;69:679–687; DOI 10.1099/ijsem.0.003200 or similar emendations in IJSEM. This means the form should be:
Emended description of Cutibacterium modestum Dekio et al. 2020
Cutibacterium modestum ( mo. des ́tum. L. neut. adj. modestum, moderate, referring to its relatively modest type of metabolism).
…The text of the emendation
The type strain is M12T (JCM 33380T =DSM 109769T), other strains are …
The quality of table 1 should be improved as the resolution is low and some letters are clipped.
Write leucylglycine arylamidase, not leucilglycine arylamidase
Line 248: replace estimate by identify
Line 250: this sentence is odd; maybe deletion of we would help.
Line 286: close to
Lines 293: please rephrase the sentence
Reviewer 2 Report
In this article, the authors studied the genetic and biochemical characteristics of eight strains belonging to the species Cutibacterium modestum and previously named Propionibacterium humerusi. The study confirmed that these strains belong to C. modestum.
The article is well writed and presented. I have no specific comment about this work.
Reviewer 3 Report
Itaru Dekio et al. characterize ng isolates of Cutibacterium modestum. The work uses a multidisciplinary approach to show low heterogeneity among isolates. However, the manuscript lacks details in the analyses. The sampling strategy may influence the analyses, as low heterogeneity may reflect the close phylogenetic proximity among isolates. These limitations have to be extensively discussed in the manuscript, particularly when the results are compared with other species.
L28: “digital comparison” is a strange phrasing.
L37: “intra-species variation of C. modestum is relatively small compared with that of C. Acnes” – this comparison is risky, as sampling may be very different from both species. The authors would require the same sampling strategy (size, locations, etc) to be able to draw such conclusion.
L100: The Methods require more details. For example, how was the DNA sequence in GridION. Provide a detailed protocol.
L107: more details are necessary for the phylogenetic analysis. Which parameters were used?
Figure 2: the tree lacks statistical support. I recommend the use of a ML or Bayes tree.
Figure 3: are the polymorphisms in coding regions? If so, do they change amino acids?
Figure 3: Details on the alignment program are necessary.
L278: “didn’t” -avoid contractions
L318: one polymorphism is not enough to draw any relevant conclusion about structuring and strain differences.
Reviewer 4 Report
The article deals with interesting issues, the methodology used in the experiment covers the most modern methods of bacterial identification and differentiation. However, I have a few substantive doubts:
1.The title of the article "Low heterogeneity among isolates of Cutibacterium modestum, a resident of human skin with a possible infectious nature, and emended descriptions of the species" indicates a completely different content. I propose to adjust it to the issues discussed in it, eg the use of the phrase "resident of human skin" indicates issues related to the medical approach to the subject, and the content of these issues is not particularly developed.
2. The "Introduction" chapter should be expanded because it lacks the characteristics and meaning of the analyzed microorganisms.
3. The purpose of the research is not very clear (for me), so please clarify it.
4. I do not really understand the content of "Conclusions", which are not clearly related to the views presented in the paper, therefore, they should be reformulated so that after reading you can summarize the content of the article concisely.
5. Has the research presented in the article been repeated several times? Or was it not anticipated repetitions in the experiment?
